# Marcus Cross-Relationship Probed by Time-Resolved CIDNP

**DOI:** 10.3390/ijms241813860

**Published:** 2023-09-08

**Authors:** Maksim P. Geniman, Olga B. Morozova, Nikita N. Lukzen, Günter Grampp, Alexandra V. Yurkovskaya

**Affiliations:** 1International Tomography Center SB RAS, 630090 Novosibirsk, Russia; m.geniman@g.nsu.ru (M.P.G.); om@tomo.nsc.ru (O.B.M.); luk@tomo.nsc.ru (N.N.L.); 2Department of Natural Sciences, Novosibirsk State University, 630090 Novosibirsk, Russia; 3Institute of Physical and Theoretical Chemistry, Graz University of Technology, Stremayrgasse, 9, A-8010 Graz, Austria; grampp@tugraz.at

**Keywords:** chemically induced nuclear polarization (CIDNP), guanosine monophosphate, tyrosine anion, short-lived radicals, degenerate electronic exchange, Marcus theory

## Abstract

The time-resolved CIDNP method can provide information about degenerate exchange reactions (DEEs) involving short-lived radicals. In the temperature range from 8 to 65 °C, the DEE reactions of the guanosine-5′-monophosphate anion GMP(-H)^−^ with the neutral radical GMP(-H)^•^, of the N-acetyl tyrosine anion N-AcTyrO^−^ with a neutral radical N-AcTyrO^•^, and of the tyrosine anion TyrO^−^ with a neutral radical TyrO^•^ were studied. In all the studied cases, the radicals were formed in the reaction of quenching triplet 2,2′-dipyridyl. The reorganization energies were obtained from Arrhenius plots. The rate constant of the reductive electron transfer reaction in the pair GMP(-H)^•^/TyrO^−^ was determined at T = 25 °C. Rate constants of the GMP(-H)^•^ radical reduction reactions with TyrO^−^ and N-AcTyrO^−^ anions calculated by the Marcus cross-relation differ from the experimental ones by two orders of magnitude. The rate constants of several other electron transfer reactions involving GMP(-H)^−^/GMP(-H)^•^, N-AcTyrO^−^/N-AcTyrO^•^, and TyrO^−^/TyrO^•^ pairs calculated by cross-relation agree well with the experimental values. The rate of nuclear paramagnetic relaxation was found for the 3,5 and β-protons of TyrO^•^ and N-AcTyrO^•^, the 8-proton of GMP(-H)^•^, and the 3,4-protons of DPH^•^ at each temperature. In all cases, the dependences of the rate of nuclear paramagnetic relaxation on temperature are described by the Arrhenius dependence.

## 1. Introduction

The kinetics of electron transfer reactions are described by Marcus theory [1]. According to this theory, the rate constant of the electron transfer reaction is expressed as
(1)$$k_{12}=A_{12}exp\left(-\frac{{\left(\lambda +{\Delta G}^{0}\right)}^{2}}{4\lambda RT}\right)$$
where λ is the reorganization energy, and ΔG^0^ is the driving force of the reaction.

The electron transfer reaction is called degenerate if the donor and acceptor differ by only one electron. Degenerate reactions of electron exchange constitute a special section of Marcus theory, because the rate constant k_12_ and the reorganization energy λ_12_ of an arbitrary electron transfer reaction can be estimated through the equilibrium constant of the cross-reaction K_12_ and the rate constants k_11_, k_22_ and the reorganization energies λ_11_, λ_22_ of the corresponding degenerate electron exchange (DEE) reactions [1]:(2)$$\lambda_{12}=\frac{\lambda_{11}+\lambda_{22}}{2}$$
(3)$$k_{12}=\sqrt{k_{11}k_{22}K_{12}f_{12}}$$
(4)$$ln\left(f_{12}\right)=\frac{ln^{2}\left(K_{12}\right)}{4ln\left(\frac{k_{11}k_{22}}{A_{11}A_{22}}\right)}$$
where A_11_, A_22_ are the pre-exponents of the DEE rate constants. If both reactants are charged, the terms should be added to Equations (3) and (4) to account for the work of bringing the charged particles closer together.

Equations (2)–(4) are used to estimate the DEE rate constants if the cross-reaction rate constant and one of the DEE rate constants are known [2,3,4]. When the rate constants are known only at a single temperature, then A_11_ = A_22_ = 10^11^ M^−1^s^−1^ is often assumed, which introduces an error in the value being determined.

The purpose of this work is to probe the applicability of the Marcus cross-relation, with better accuracy provided through knowledge of the temperature dependences of the corresponding DEE rate constants. In this case, the value of the rate constant of reductive electron transfer to short-lived radicals calculated from the cross-relation does not contain errors due to the uncertainty of the pre-exponent values of the DEE rate constants, since the latter are determined from the temperature dependences of the rates.

To test the Marcus cross-relation, we chose the reduction reactions of the short-lived radicals of guanosine-5′-monophosphate, GMP(-H)^•^, by tyrosine (TyrO^−^) and N-acetyl tyrosine (N-AcTyrO^−^) anions. The DEE reactions cannot be studied using optical methods, since there is no change of reactant concentration in course of the reaction; thus, the total optical density does not change. The methods for studying DEE reactions between stable reactants (transitional metal complexes in different oxidation states; long-lived radicals with the corresponding diamagnetic molecules) usually employ modification one of the reactants. Such methods include the use of deuterated ligands [5], optical isomers [6], and radioactive isotopes [7]. These methods are not suitable for studying DEE involving short-lived radicals. The DEE reactions between stable reagents can also be studied using line broadening in the EPR spectrum [8,9,10,11] or in NMR spectra [12,13,14]. These methods are also not applicable to short-lived radicals, where the lifetime of the radicals is so short that it is impossible to record their EPR or NMR spectra under normal conditions.

The photo-induced time-resolved chemical-induced nuclear polarization (tr-CIDNP) method [15,16,17,18,19,20,21,22,23] fortunately makes it possible to study degenerate exchange reactions involving short-lived radical intermediates [24,25,26]. It is an indirect method of radical intermediate detection using short laser pulses via quantitative analysis of anomalous enhancement or emission of NMR lines of the diamagnetic reaction products and their dependence on time after a laser pulse.

The reduction reaction of the neutral radical GMP(-H)^•^ by the anions of tyrosine TyrO^−^ and N-acetyltyrosine N-AcTyrO^−^ was studied using the tr-CIDNP method; dipyridyl (DP) was used as a dye. The CIDNP kinetics for the systems GMP + DP [27], N-AcTyrOH + DP [28], and GMP + N-AcTyrOH + DP [29] were previously studied at t = 25 °C in a wide pH range. The temperature dependence for the DEE rate constant in the GMPH^+^/GMPH^•++^ pair was also measured previously [30]. In the temperature range of 8–65 °C we measured the DEE rate constants in the pairs GMP(-H)^−^/GMP(-H)^•^, TyrO^−^/TyrO^•^, and N-AcTyrO^−^/N-AcTyrO^•^. From these temperature dependences of the rate constants, the reorganization energies were determined; with use of these values and the Marcus cross-relation, the rate constants of the GMP(-H)^•^ radical reduction reaction by TyrO^−^ and N-AcTyrO^−^ anions were calculated. We compared the calculated rate constants with the electron transfer reaction rate constant for GMP(-H)^•^ + TyrO^−^ as measured at t = 25 °C and the previously determined reaction rate constant for the reactants GMP(-H)^•^ + N-AcTyrO^−^ [29]. The structures of the studied diamagnetic particles GMP(-H)^−^, N-AcTyrO^−^, TyrO^−^, and DP as well as the previously studied GMPH^+^ are shown in Figure 1.

## 2. Results

### 2.1. Mechanism of the CIDNP Effect

The CIDNP spectra recorded during irradiation of solutions containing DP + GMP(-H)^−^ and DP + TyrO^−^ with signal attribution are shown in Figure 2. The spectra were recorded with zero delay after the laser pulse, without an internal standard; the dependence of water chemical shifts on temperature was used δ(HDO) = 5.060 − 0.0122t + (2.11 × 10^−5^)t^2^ (t in °C) [31]) to relate the chemical shifts.

The CIDNP signal originates as follows. The dye molecule D absorbs a quantum of light and transitions from the ground singlet state S_0_ to the excited singlet state S_1_. Due to intersystem crossing (ISC), the S_1_ molecule state converts to the triplet electronic state. Then, as a result of diffusion motion, the dye molecule D in the triplet state encounters the quencher Q, which quenches the dye triplet state by means of electron transfer with a rate constant k_q_. The radical pair formed in the course of this reaction preserves the triplet electron state of the precursor. At the instant of their formation, the radicals are situated in the solvent “cage”. They cannot react back to the ground diamagnetic states due to the conservation of the total electron spin. In order for this to happen, the radical pair must change to the singlet state. Such a transition from the triplet to singlet state (and vice versa) can happen due to the difference of the Larmor frequencies of the radicals in the field of the NMR spectrometer (at different g-factors) and due to hyperfine couplings of electron spins of the radicals with magnetic nuclei. Thus, in the high field of the NMR spectrometer, the rate of singlet–triplet transitions depends, in addition to the g-factor difference, on the nuclear spin projections along the magnetic field and the configuration of nuclei spins (S-T_0_ mechanism of CIDNP formation in strong magnetic fields). Therefore, the in-cage, geminate, recombination products are enriched in those nuclear spin states that have a higher singlet–triplet interconversion rate. However, the total polarization of nuclear spins does not change, since the geminate reaction gives rise only to the sorting between the nuclear polarization of geminate diamagnetic products and the polarization of the radicals that escaped into the bulk of the solution. Consequently, at the initial instant of time (on the scale of T_1_ nuclear relaxation time in radicals), the nuclear polarization of the radicals that escaped into the bulk of the solution is equal in magnitude and opposite in sign to the polarization of the diamagnetic products formed in the course of geminate recombination.

The sign of CIDNP signal Г of geminate diamagnetic products is determined by the Kaptein rule [32], Г = μ × sgn(Δg) × sgn(A), where μ = +1 in the case of a triplet precursor and μ = −1 in the case of a singlet one; sgn(Δg) is the sign of the g-factor difference of the radicals; sgn(A) is the sign of the hfi-coupling constant of the corresponding nucleus. The value of the g-factor of the neutral GMP(-H)^•^ radical is equal to 2.0034 [33], and the g-factor for tyrosine radicals is 2.0041 [34,35]. The sign of the hfi-coupling constant of the H-8 proton of the GMP(-H)^•^ radical is negative [33]; for TyrO^•^ and N-AcTyrO^•^, the sign of the hfi-coupling constants of the H-3,5 protons is also negative, and of the H-2,6, β protons it is positive [34,35]. There are no experimental data for the DPH^•^ radical, the calculated hfi-coupling constants of the H-3,4,5,6 protons are negative [36], and the value of the g-factor of the DP^−•^ radical anion is equal to 2.0030 [37]. Kaptein’s rule holds for all signals in all ^1^H CIDNP spectra obtained in this work.

### 2.2. Experimental Data Processing

Theoretical fitting of the CIDNP kinetics measured for the systems GMP(-H)^−^ + DP, N-AcTyrO^−^ + DP, and TyrO^−^ + DP was performed using the Fisher model [38]. The model takes into account the nuclear polarization transfer from radical into diamagnetic molecules due to the second-order radical recombination, paramagnetic nuclear relaxation in radicals, the arising of polarization within re-contacts of radicals in the bulk, and polarization transfer from radical to molecule as a result of DEE. The system of Equations (5)–(7) describes the time evolution of the radical pair concentration R(t), the nuclear polarization in radicals P_R_(t), and the experimentally registered nuclear polarization in diamagnetic molecules P_Pr_(t):(5)$$R\left(t\right)=\frac{R_{0}}{1+{k_{R}R}_{0}t}$$
(6)$$\frac{dP_{R}\left(t\right)}{dt}=-k_{R}R(t)P_{R}(t)-k_{R}{\beta R}^{2}(t)-k_{obs}C_{q}P_{R}(t)-\frac{P_{R}(t)}{T_{1}}$$
(7)$$\frac{dP_{Pr}\left(t\right)}{dt}=k_{R}R(t)P_{R}(t)+k_{R}\beta R^{2}(t)+k_{obs}C_{q}P_{R}\left(t\right)$$
with the initial condition of P_Pr_(t = 0) = −P_R_(t = 0) = P_G_.

Here, R_0_ is the initial concentration of radical pairs; k_R_ is the rate constant for the second-order radical recombination; T_1_ is the nuclear spin relaxation time in radicals; k_obs_ is the rate constant for DEE; C_q_ is the concentration of diamagnetic quencher molecules; β = γP_G_/R_0_, showing the amount of polarization arising in one secondary radical pair (the radical pairs formed by radicals escaped from different geminate pairs, or so-called F-pairs); γ is the ratio of CIDNP polarization formed in the secondary radical pair to the polarization in the geminate pair (in the case of a triplet precursor, γ ≈ 3; usually γ = 2.8 is taken [27]); P_G_ is the geminate polarization.

The system of Equations (5)–(7) is written assuming that the fraction of geminate recombination of radical pairs is negligible and that the radical formation due to quenching of the triplet state of the dye occurs instantaneously. Equation (5) describes the time dependence of the radical concentration, which decreases according to the second-order recombination kinetics. The first terms in Equations (6) and (7) describe the transfer of polarization from radicals to diamagnetic molecules due to the recombination of radicals, and the third terms in both Equations (6) and (7) describe the transfer of polarization from radicals to diamagnetic molecules due to the DEE process. The second terms in Equations (6) and (7) describe the polarization forming as a result of radicals encountered in the bulk, i.e., in F-pairs. The fourth term in Equation (6) describes the decay of nuclear polarization in radicals as a result of nuclear paramagnetic relaxation.

Under experimental conditions, there are DPH^•^/DP pairs (pKa(DPH^•^) > 14, [39]) in which there is no DEE. Therefore, when modeling the kinetic curves for nuclear polarization of DP, we took k_obs_ = 0.

In the system GMP(-H)^−^ + TyrO^−^ + DP, the kinetics of nuclear polarization are also affected by the reduction reaction of the GMP(-H)^•^ radical by the TyrO^−^ anion. Then, the time evolution of the concentrations of GMP(-H)^•^, TyrO^•^, and DPH^•^ radicals and the polarizations of the nuclei in P_R_^GMP^ and P_R_^Tyr^ radicals and P_Pr_^GMP^ and P_Pr_^Tyr^ diamagnetic molecules are described by the following equations:(8)$$\frac{dR^{Tyr}}{dt}=-k_{R}^{Tyr}R^{Tyr}R^{DP}+k_{obs}^{red}R^{GMP}C^{Tyr}$$
(9)$$\frac{dR^{GMP}}{dt}=-k_{R}^{GMP}R^{GMP}R^{DP}-k_{obs}^{red}R^{GMP}C^{Tyr}$$
(10)$$R^{DP}=R^{GMP}+R^{Tyr}$$
(11)$$\frac{dP_{R}^{GMP}}{dt}=-k_{R}^{GMP}R^{DP}P_{R}^{GMP}-k_{R}^{GMP}\beta^{GMP}R^{DP}R^{GMP}-k_{obs}^{GMP}C^{GMP}P_{R}^{GMP}-k_{obs}^{red}P_{R}^{GMP}C^{Tyr}-\frac{P_{R}^{GMP}}{T_{1}^{GMP}}$$
(12)$$\frac{dP_{Pr}^{GMP}}{dt}=k_{R}^{GMP}R^{DP}P_{R}^{GMP}+k_{R}^{GMP}\beta^{GMP}R^{DP}R^{GMP}+k_{obs}^{GMP}C^{GMP}P_{R}^{GMP}+k_{obs}^{red}P_{R}^{GMP}C^{Tyr}$$
(13)$$\frac{\begin{array}{c} dP_{R}^{Tyr} \end{array}}{dt}=-k_{R}^{Tyr}R^{DP}P_{R}^{Tyr}-k_{R}^{Tyr}\beta^{Tyr}R^{DP}R^{Tyr}-k_{obs}^{Tyr}C^{Tyr}P_{R}^{Tyr}-\frac{P_{R}^{Tyr}}{T_{1}^{Tyr}}$$
(14)$$\frac{dP_{Pr}^{Tyr}}{dt}=k_{R}^{Tyr}R^{DP}P_{R}^{Tyr}+k_{R}^{Tyr}\beta^{Tyr}R^{DP}R^{Tyr}+k_{obs}^{Tyr}C^{Tyr}P_{R}^{Tyr}$$
with initial conditions P_Pr_^GMP^(t = 0) = −P_R_^GMP^(t = 0) = P_G_^GMP^ and P_Pr_^Tyr^(t = 0) = −P_R_^Tyr^(t = 0) = P_G_^Tyr^, where k_R_^GMP^ and k_R_^Tyr^ are the recombination rate constants of the radicals DPH^•^ with GMP(-H)^•^ and TyrO^•^ radicals, respectively; k_obs_^red^ is the observed rate constant of reduction of the GMP(-H)^•^ radical by the tyrosinate-anion; β^GMP^ = γP_G_^GMP^/R_0_^GMP^ and β^Tyr^ = γP_G_^Tyr^/R_0_^Tyr^.

In the process of solving this system of equations, the parameters k_R_^GMP^/k_R_^Tyr^ and k_q_^GMP^/k_q_^Tyr^ are introduced; through the latter, the ratio of initial concentrations of GMP(-H)^•^ and TyrO^•^ radicals is expressed.

The initial concentration of radical pairs R_0_ is proportional to the fraction of light absorbed by the dye. The TyrO^−^ and N-AcTyrO^−^ compounds absorb at λ = 308 nm; as the concentration of tyrosine in the solution increases, the fraction of light absorbed by DP decreases, and accordingly, the initial concentration of radical pairs decreases as well. The ratio of initial concentrations of radical pairs in two experiments with different concentrations of tyrosine is as follows:(15)$$\frac{R_{0}\left(C_{1}^{Tyr}\right)}{R_{0}\left(C_{2}^{Tyr}\right)}=\frac{\varepsilon^{DP}C^{DP}+\varepsilon^{Tyr}C_{2}^{Tyr}}{\varepsilon^{DP}C^{DP}+\varepsilon^{Tyr}C_{1}^{Tyr}}$$

In Equation (15) ε^DP^ and ε^Tyr^ are the extinction coefficients at λ = 308 nm; C^DP^ is the DP concentration; C_1_^Tyr^ and C_2_^Tyr^ are tyrosine concentrations; ε^DP^ = 1.2 × 10^3^ M^−1^cm^−1^ [40] and ε^TyrO−^ = ε^N−AcTyrO−^ = 790 ± 8 M^−1^cm^−1^ were determined in this work. This effect was taken into account when modeling CIDNP kinetics in systems containing TyrO^−^ and N-AcTyrO^−^.

The duration of the RF pulse (1–2 μs) is comparable with the characteristic time of the CIDNP decay, so it is necessary to take into account the polarization evolution during the RF pulse. Consideration of the CIDNP kinetics during the RF pulse for the special case of an ideal rectangular pulse is considered in [41]; the case of an RF pulse of arbitrary shape is considered in [42]. A similar approach was used in the works of Morozova and Yurkovskaya [43,44,45,46,47,48,49,50].

To take into account the CIDNP kinetics during the application of the RF pulse along the *x*-axis, we considered the following system of equations, which takes into account the polarization of the nuclei along the *y*- and *z*-axes:(16)$$\frac{dP_{Pr}^{z}}{dt}=\frac{dP_{Pr}}{dt}-w\left(t\right)P_{Pr}^{y}$$
(17)$$\frac{dP_{Pr}^{y}}{dt}=w\left(t\right)P_{Pr}^{z}$$
where w(t) is the pulse shape; the equation for dP_Pr_/dt is given by Equation (7); before the pulse, $P_{Pr}^{y}$ = 0. The signal observed in the experiment is proportional to the after-pulse value of $P_{Pr}^{y}$. The expression for polarization along the *y*-axis after the pulse is as follows [41]:(18)$$P_{Pr}^{y}=\int_{0}^{T}P_{Pr}\left(t_{0}+t\right)w\left(t\right)cos\left(\int_{t}^{T}w\left(z\right)dz\right)dt$$
where t_0_ is the start time of the RF pulse, and T is its duration.

The duration of the π pulse is equal to t_π_ = 12.70 µs. The shapes of RF pulses of 1 and 2 µs duration used in this work were determined using an oscilloscope.

### 2.3. Results of CIDNP Kinetics Treatment

In the GMP(-H)^−^ + DP system, the polarization time dependences were modeled for the 8th GMP proton and 3,4 DP protons. In the TyrO^−^ + DP and N-AcTyrO^−^ + DP systems, the dependences were modeled for the 3,5 and β-protons of tyrosine and 3,4 protons of DP—the other signals had too-low signal-to-noise ratios. In the GMP(-H)^−^ + TyrO^−^ + DP system, CIDNP polarization time dependences were modeled for 8th GMP proton and 3,5 tyrosine protons. Values of the degenerate electron exchange rate constants and paramagnetic nuclear relaxation times obtained from the best agreement between the calculated and experimental curves are given in Table 1, Table 2 and Table 3; Figure 3a–c shows examples of simulations of these experimental data.

For the GMP(-H)^−^ + DP, we did not determine T_1_ for 3,4 protons of DP at 65 °C since perdeuterated d_8_-DP was used to avoid the overlap of the ^1^H NMR signals of H-8 of GMP and H-3,4 of DP, which occurs at this temperature. For N-AcTyrO^−^ + DP at 65 °C and TyrO^−^ + DP at 45 °C, we failed to determine T_1_ for H-β.

For GMP(-H)^−^ + DP at 25 °C, the value of the DEE rate constant differs from the literature data, while the paramagnetic relaxation time of the 8th proton in the GMP(-H)^•^ radical coincides well with the literature data: k_obs_ = 4.0 × 10^7^ M^−1^s^−1^, T_1_ = 20 μs [27]. Moreover, the T_1_ for the 8th proton in the GMP(-H)^•^ radical coincides with the previously found T_1_ for the 8th proton in the GMPH^•++^ radical over the entire temperature range [30]. For the N-AcTyrO^−^ + DP at t = 25 °C, the values of the DEE rate constant and paramagnetic relaxation time of 3,5 protons coincide with the literature data: k_obs_ = 6.0 × 10^7^ M^−1^s^−1^, T_1_ = 63 µs [29]; k_obs_ = 4 × 10^7^ M^−1^s^−1^, T_1_ = 60 µs [51]. However, the best fit values of nuclear paramagnetic relaxation times for 3,4 protons of the DPH^•^ radical do not coincide with the literature data for T_1_ = 44 µs [28,40] and T_1_ = 45 µs [51].

Simulation of the experimental data for DP + GMP(-H)^−^ + TyrO^−^ is shown in Figure 3d; the parameters found are as follows: k_obs_^red^ = 1.5 × 10^8^ M^−1^s^−1^, k_q_^GMP^/k_q_^Tyr^ = 0.3, and k_R_^GMP^/k_R_^Tyr^ = 1.3. The parameters k_obs_^GMP^, k_obs_^Tyr^, T_1_^GMP^, and T_1_^Tyr^ found in this work were used in the calculation. There are literature data for the reduction of the GMP(-H)^•^ radical by the N-AcTyrO^-^ anion: k_obs_^red^ = 1.6 × 10^8^ M^−1^s^−1^, k_q_^GMP^/k_q_^Tyr^ = 0.56, and k_R_^GMP^/k_R_^Tyr^ = 0.9 [29].

### 2.4. Temperature Dependence of the DEE Rate Constants

For the estimation of the effect of diffusion, DEE kinetic scheme (Figure 1) was considered (using tyrosine as an example).

In Figure 1, the first-order DEE rate w_ex_ is related to the second-order rate constant k_et_ as follows:(19)$$w_{ex}=k_{et}\frac{k_{-d}}{k_{d}}$$

k_obs_ is the observable DEE rate constant; k_obs_ is expressed through k_et_ and k_d_ as follows:(20)$$k_{obs}=\frac{k_{et}k_{d}}{2k_{et}+k_{d}}$$

The diffusion rate constant k_d_ was estimated by the Smoluchowski equation, taking into account the charges of the reactants [52,53]:(21)$$k_{d}=\frac{2}{3\eta }\cdot \frac{{\left(r_{1}+r_{2}\right)}^{2}}{r_{1}r_{2}}\cdot \frac{w_{r}}{exp\left(\frac{w_{r}}{RT}\right)-1}$$
where r_1_ and r_2_ are the radii of the reagents; $w_{r}$ is the work function of approach-charged reactants to each other:(22)$$w_{r}=\frac{z_{1}z_{2}e^{2}N_{A}}{4\pi \varepsilon_{0}\varepsilon \left(r_{1}+r_{2}\right)}f$$

We assume that the reactants approach to a distance equal to the sum of their radii. The factor f takes into account the ionic strength of the solution:(23)$$f={\left(1+\left(r_{1}+r_{2}\right)B\sqrt{\frac{\mu }{\varepsilon T}}\right)}^{-1}$$
where B = (2N_A_e^2^/(ε_0_k))^1/2^ = 50.345 l^1/2^K^1/2^Å^−1^M^−1/2^.

The viscosity values of D_2_O were taken from [54,55], and dielectric permittivity from [56]. The average elliptical radii of the reactants were used to estimate r_1_ and r_2_. An ellipsoid of the smallest volume was described around the molecule, and the radius was expressed as follows [57]:(24)$$\frac{1}{r}=\frac{F\left(\varphi ,\alpha \right)}{\sqrt{\left(a^{2}-c^{2}\right)}}$$
where F(φ,α) is an elliptic integral of the first kind; φ = arcsin((a^2^ − c^2^)^1/2^/a); α = ((a^2^ − c^2^)/(a^2^ − b^2^))^1/2^; a, b, and c are the semiaxes of the ellipsoid, and a ≥ b ≥ c.

The ellipsoid semiaxes were calculated using the N. Shor algorithm [58] with the molecular geometries of the crystal structures of DL-tyrosine [59], N-acetyl-L-tyrosine, and guanosine-5′-monophosphate trihydrate [60]. The molecular radii found were r(GMP(-H)^−^) = 4.18 Å, r(N-AcTyrO^−^) = 3.82 Å, and r(TyrO^−^) = 3.06 Å; we assumed that the radii of the radicals do not differ from those of the corresponding diamagnetic particles.

When calculating w_r_, the negatively charged carboxyl and phosphate groups in the reactants were taken into account: z(GMP(-H)^−^) = −3, and z(TyrO^−^) = −2.

The k_et_ values calculated by Equation (20) are given in the last columns of Table 1, Table 2 and Table 3.

The first-order DEE rate constant w_ex_ depends on temperature, as follows [1]:(25)$$w_{ex}\left(T\right)=\frac{k_{0}}{\sqrt{T}}exp\left(-\frac{\lambda }{4RT}\right)$$

The second-order DEE rate constant k_et_ is expressed as follows:(26)$$k_{et}=\frac{k_{d}}{k_{-d}}k_{ex}=K_{A}w_{ex}$$

The K_A_ in Equation (26) is the equilibrium constant of the pre-reaction complex formation; it depends on temperature, as follows [61]:(27)$$K_{A}=4\pi N_{A}{\left(r_{1}+r_{2}\right)}^{2}\delta d\cdot exp\left(-\frac{w_{r}}{RT}\right)=K_{0}exp\left(-\frac{w_{r}}{RT}\right)$$
where δd is the reaction zone thickness. Then, the second-order rate constant k_et_ depends on temperature, as follows:(28)$$k_{et}=\frac{K_{0}k_{0}}{\sqrt{T}}exp\left(-\frac{w_{r}}{RT}-\frac{\lambda }{4RT}\right)=\frac{A}{\sqrt{T}}exp\left(-\frac{w_{r}}{RT}-\frac{\lambda }{4RT}\right)$$

The work function $w_{r}$ depends on the temperature through dependence on temperature of the D_2_O dielectric permittivity, at each temperature $w_{r}$; this can be calculated by Equation (22). To find the pre-exponent A and the reorganization energy λ, we plot the values of ln[k_et_√Texp($w_{r}$/RT)], where the k_et_ values are experimental ones, versus 1/T (Figure 4), and approximated these points by linear dependence. The values of A and λ found from the approximation are given in Table 4.

Previously, we studied [30] the temperature dependence of the DEE rate constant between the GMPH^+^ cation and the GMPH^•++^ dication radical; the reorganization energy found for this reaction is λ = 0.79 ± 0.11 eV [30] and coincides with the value for GMP(-H)^−^ + GMP(-H)^•^ (see Table 4).

### 2.5. The Marcus Cross-Relation Equation

To account for the effects of diffusion, the kinetic scheme of electron transfer (Figure 2) was considered (for the radical GMP(-H)^•^ reduction by anion TyrO^−^, as an example).

In Figure 2, the first-order rate w_12_ is related to the second-order rate constant k_12_, as follows:(29)$$w^{12}=\frac{k_{-d}^{12}}{k_{d}^{12}}k^{12}$$

The observable second-order rate constant $k_{obs}^{12}$ is expressed through $k^{12}$ and $k_{d}^{12}$, as follows:(30)$$k_{obs}^{12}=\frac{k^{12}k_{d}^{12}}{k^{12}+k_{d}^{12}}$$

The rate constants $k^{12}$(TyrO^−^) and $k^{12}$(N-AcTyrO^−^) of the GMP(-H)^•^ reduction by TyrO^−^ and N-AcTyrO^−^, respectively, after correction for diffusion (k_d_ was calculated according Equation (21)) are the following: $k^{12}$(TyrO^−^) = 1.7 × 10^8^ M^−1^s^−1^ and $k^{12}$(N-AcTyrO^−^) = 1.8 × 10^8^ M^−1^s^−1^ [29]. Thus, the values of the reduction rate constants are very close.

The rate constant $w^{12}\;$depends on the temperature, as follows:(31)$$w^{12}=\frac{k_{0}}{\sqrt{T}}exp\left(-\frac{{\left({\Delta G}_{12}^{0’}+\lambda_{12}\right)}^{2}}{4RT\lambda_{12}}\right)$$
where ${\Delta G}_{12}^{0’}\;$is the driving force of the first-order redox reaction.

The equilibrium constant between the initial reactants and reaction products, $K_{12},\;$is expressed as follows:(32)$$K_{12}=exp\left(-\frac{{\Delta G}_{12}^{0}}{RT}\right)=K_{A}^{12}K_{12}\prime {\left(K_{A}^{21}\right)}^{-1}=exp\left(\frac{-{\Delta G}_{12}^{0^{’}}-w_{12}+w_{21}}{RT}\right)$$
where $K_{A}^{12}$ and $K_{A}^{21}$ are the equilibrium constants of the formation of pre-reaction and post-reaction complexes, expressed by Equation (27). Assuming the radii of radicals and corresponding diamagnetic particles are the same, the pre-exponents in Equation (32), the factors $K_{A}^{12}$ and $K_{A}^{21}$, can be shortened (reduced). The ${\Delta G}_{12}^{0}$ value is expressed through standard electrode potentials; thus, ${\Delta G}_{12}^{0^{\prime }}$ can be calculated.

The expression for the second-order electron transfer rate constant $k^{12}$ is given as follows:(33)$$k^{12}=K_{A}^{12}w^{12}=\frac{A_{12}}{\sqrt{T}}exp\left(-\frac{w_{12}}{RT}\right)exp\left(-\frac{{\left({\Delta G}_{12}^{0}-w_{12}+w_{21}+\lambda_{12}\right)}^{2}}{4RT\lambda_{12}}\right)$$

If the electron transfer occurs over a relatively long distance with a weak force interaction between the reactants, then one can assume that λ_12_ = (λ_11_ + λ_22_)/2, that is, the Marcus cross-relation, where λ_11_ and λ_22_ are the reorganization energies for the respective DEE reactions.

If we additionally assume the following ratio between the pre-exponents, $A_{12}$ = ($A_{11}A_{22}$)^1/2^, then we can express $k^{12}$ through the temperature dependence of parameters of the DEE rate constants. Thus, the cross-reaction rate constant $k_{calc}^{12}$ was calculated employing the following formula:(34)$$k_{calc}^{12}=\frac{\sqrt{A_{11}A_{22}}}{\sqrt{T}}exp\left(-\frac{w_{12}}{RT}\right)exp\left(-\frac{{\left({\Delta G}_{12}^{0}-w_{12}+w_{21}+\left(\lambda_{11}+\lambda_{22}\right)/2\right)}^{2}}{2RT\left(\lambda_{11}+\lambda_{22}\right)}\right)$$

#### Reactant Parameters for the Marcus Cross-Relation Calculations

The cross-relation calculations require the values of the difference of standard electrode potentials, the reorganization energy, and the pre-exponent for the corresponding DEE reactions, as well as the radii of all reactants and products to calculate the work function. In all the cases, we assume that the radii of the radicals and the corresponding diamagnetic particles are the same.

If the DEE rate constant is only known at a certain temperature, we estimate the pre-exponent to be A_11_ = 10^11^ M^−1^s^−1^ and calculate the reorganization energy using Equation (35).
(35)$$\lambda_{11}=-4w_{r}-4RT\times ln\left(\frac{k_{11}\sqrt{T}}{A_{11}}\right)$$

All the parameters of the reactants are given in Table 5. The Z_red_ column shows the charge of the reduced form of each reactant. The structures of some reactants are shown in Figure 5.

The TyrO^•^/TyrO^−^ case. The standard electrode potential is E^0^ = 0.72 V [62]. The radius calculation is described above; the pre-exponent and reorganization energy for the DEE reaction were found experimentally in this work.

The N-AcTyrO^•^/N-AcTyrO^−^ case. We use the value of the standard electrode potential for N-acetyltyrosine methyl ester at pH = 7, E_7_ = 0.97 V [63]. If pK_a_(N-AcTyrOH) = 10.1, then E^0^ = 0.79 V. The calculation of the radius is described above; the pre-exponent and reorganization energy for the DEE reaction were found experimentally in this work.

The GMP(-H)^•^/GMP(-H)^−^ case. The standard electrode potential of GMP at pH = 7, E_7_ = 1.31 V [64], determined from photoinduced electron transfer reaction rates, agrees well with that of guanosine at pH = 7, E_7_ = 1.29 V [65], determined by pulsed radiolysis. The acidity constants of GMP [66] and its radical [67,68,69] are shown in Figure 3. For GMP(-H)^•^/GMP(-H)^−^, the calculated potential is E^0^ = 1.23 V.

The GMPH^•++^/GMPH^+^ case. The standard electrode potential E^0^ = 1.55 V is calculated from E_7_ = 1.29 V [65]. The pre-exponent and reorganization energies for the DEE reaction were found earlier [30]. The average elliptical radius, r = 4.19 Å, was calculated using the method described above; the geometry was taken from the crystal structure of guanosine-5′-monophosphate trihydrate [60].

The N-AcTrpH^+•^/N-AcTrpH case. We use the value of the standard electrode potential for tryptophan, E^0^ = 1.15 V [62]. The DEE rate constant at 25 °C, k_11_ = 9 × 10^8^ M^−1^s^−1^, was determined experimentally by CIDNP with microsecond resolution [40]. After correction for diffusion using Equation (20), one obtains k_11_ = 1.3 × 10^9^ M^−1^s^−1^. The mean elliptical radius, r = 3.83 Å, was calculated using the method described above; the geometry was taken from the N-AcTrpH crystal structure [70]. We assume that A_11_/√T = 10^11^ M^−1^s^−1^; then, λ = 0.45 eV.

The ClO_2_^•^/ClO_2_^−^ case. The standard electrode potential is E^0^ = 0.934 V [71]. The DEE constant at 25 °C is k_11_ = 3.3 × 10^4^ M^−1^s^−1^, determined from a cross-relation with parameters A_11_/√T = 10^11^ M^−1^s^−1^ and r = 1.5 Å [72]. Then, λ_11_ = 1.53 eV.

The N_3_^•^/N_3_^−^ case. The standard electrode potential is E^0^ = 1.33 V [71]. The DEE rate constant at 25 °C is k_11_ = 3.7 × 10^6^ M^−1^s^−1^, determined from a cross-relation with parameters A_11_/√T = 10^11^ M^−1^s^−1^ and r = 2 Å [72]. Then, λ_11_ = 1.05 eV.

The NO_2_^•^/NO_2_^−^ case. The standard electrode potential is E^0^ = 1.04 V [71]. The DEE rate constant at 25 °C is k_11_ = 580 M^−1^s^−1^, determined experimentally by isotopic substitution [73]. We assume that A_11_/√T = 10^11^ M^−1^s^−1^; then, λ = 1.95 eV. The radius is r = 1.9 Å [72].

The [IrCl_6_]^2−^/[IrCl_6_]^3−^ case. The standard electrode potential is E^0^ = 0.892 V [74]. The DEE constant at 25 °C is k_11_ = 2.3 × 10^5^ M^−1^s^−1^, determined experimentally by isotopic substitution [75]. We assume that A_11_/√T = 10^11^ M^−1^s^−1^. The radius is r = 4.4 Å [76], and the DEE rate constant measured at μ = 0.1 M. Then, λ = 1.04 eV.

The CysS^•^/CysS^−^ case. The standard electrode potential is E^0^ = 0.76 V [77]. The DEE rate constant at 25 °C is k_11_ = 5.4 × 10^3^ M^−1^s^−1^, determined from the Marcus cross-relation with parameters A_11_/√T = 10^11^ M^−1^s^−1^ and r = 3.0 Å; the k_12_ cross-reaction rate constant was measured at μ = 0.1 M [77]. Then, λ = 1.48 eV.

The TEMPO^+^/TEMPO^•^ case. The standard electrode potential is E^0^ = 0.745 V [78]. The DEE rate constant at 25 °C is k_obs_ = 8.6 × 10^7^ M^−1^s^−1^, determined experimentally by line broadening in the EPR spectrum [9]. After correction for diffusion by Equation (20), k_11_ = 8.8 × 10^7^ M^−1^s^−1^. We assume that A_11_/√T = 10^11^ M^−1^s^−1^. Then, λ = 0.72 eV. The radius is r = 3.67 Å [9].

### 2.6. The T_1_ Temperature Dependence

The temperature dependences of the nuclear paramagnetic relaxation times, T_1_, were obtained by means of the CIDNP kinetics simulation. We assume that the main mechanism of relaxation is modulation of the hyperfine interaction (hfi) tensor due to the stochastic rotation of the molecule as a whole. Then, the nuclear paramagnetic relaxation time T_1_ should depend on the magnetic field B and the rotational correlation time of the molecule, as follows:(36)$$\frac{1}{T_{1}}\propto \frac{\tau_{M}}{1+w^{2}\tau_{M}^{2}}$$

If the diffusive stochastic rotation of the radical is fast, $\gamma^{2}B^{2}\tau_{M}^{2}\ll 1$, then the nuclear paramagnetic relaxation rate, 1/T_1_, is proportional to $\tau_{M}$; the value of $\tau_{M}$ can be estimated from the Stokes–Einstein–Debye equation:(37)$$\tau_{M}=\frac{4\pi a^{3}\eta }{3kT}$$

According to [54,55], the temperature dependence of D_2_O viscosity in the temperature range 8–65 °C is well described by the following equation:(38)$$\eta \left(T\right)=\eta_{0}exp\left(\frac{E}{RT}\right)$$
where E/R = 2059 K, and η_0_ = 1.14 × 10^−6^ Pa × s.

Then, the T_1_ temperature dependence should be described by the following equation:(39)$$T_{1}\left(T\right)=CTexp\left(-\frac{E}{RT}\right)$$
where C and E/R are parameters; E/R = 2059 K. 

The T_1_ temperature dependence from Equation (39) is linearized in the coordinates ln(T/T_1_) vs. 1/T (Figure 6).

While fitting using straight lines, we discarded by the dropout values. The values from the slopes of the straight lines, E/R, are summarized in Table 6. For 3,5 N-AcTyrO^•^, 8th GMP(-H)^•^, and H3,4 DPH^•^ protons, the slopes coincide within an error with the slope from the viscosity dependence. The slope for TyrO^•^ 3,5 protons is almost two times lower than the viscosity slope. Earlier, we found the slope for the 8th proton GMPH^•++^; it also coincides with the slope from viscosity dependence.

For the H-3,5 of N-AcTyrO^•^, H-8 of GMP(-H)^•^, and H-3,4 of DPH^•^, the slopes of the fitted straight lines coincide with the slope for viscosity within an error margin, indicating that rotational modulation of the hfi tensor is the main mechanism of nuclear paramagnetic relaxation of these protons.

The defined energy value, E, for the H-3,5 of TyrO^•^ protons is notably different from that for the viscosity dependence. This result can be rationalized assuming that for these protons, two mechanisms give rise to paramagnetic relaxation: modulation of the hfi tensor due to stochastic rotation of the molecule as a whole and modulation of the hfi tensor due to stochastic rotation of the phenolic ring. Assuming that the ring rotation has a small activation energy, the temperature dependence of the correlation time τe for this rotation is as follows:(40)$$\tau_{e}=\frac{C}{T}exp\left(\frac{E}{RT}\right)\approx \frac{C_{2}}{T}$$
while the temperature dependence of the rotational correlation time τ_M_ is as follows:(41)$$\tau_{M}=\frac{C_{1}}{T}exp\left(\frac{E}{RT}\right)$$

Then, the nuclear relaxation time T_1_ can be expressed by employing the Lipari–Szabo [79,80] equation:(42)$$\frac{1}{T_{1}}=C\times \left(S^{2}\tau_{M}+\left(1-S^{2}\right)\tau \right)$$
(43)$$\tau =\frac{\tau_{M}\tau_{e}}{\tau_{M}+\tau_{e}}=\frac{C_{1}}{T}\times \frac{exp\left(\frac{E}{RT}\right)\times exp\left(\frac{E}{RT_{h}}\right)}{exp\left(\frac{E}{RT}\right)+exp\left(\frac{E}{RT_{h}}\right)}$$
where S^2^ is an order parameter, playing the role of the dimensionless amplitude of molecular motion: the smaller the S^2^, the larger the amplitude; T_h_ is the temperature when the correlation times τ_e_ and τ_M_ coincide; C_2_ = C_1_ × exp(E/(R·T_h_).

The temperature dependence of the ln(T/T_1_) on 1/T has the following form:(44)$$ln\left(\frac{T}{T_{1}}\right)=ln\left(C\right)+\frac{E_{n}}{RT}+ln\left(S^{2}+\left(1-S^{2}\right)\times \frac{exp\left(\frac{E}{RT_{half}}\right)}{exp\left(\frac{E}{RT}\right)+exp\left(\frac{E}{RT_{half}}\right)}\right)$$
where E/R = 2059 K is the activation energy for D_2_O viscosity. There are three fit parameters: C, S^2^, and T_h_. The value S^2^ = 0.01 was taken for the fit. The best fit of the experimental data is then achieved at T_h_ = 850 K; see Figure 7.

The paramagnetic relaxation time T_1_ of the methylene protons (β-CH_2_ protons) in the N-AcTyrO^•^ and TyrO^•^ radicals is almost independent of temperature (Table 2 and Table 3). We assume that the paramagnetic relaxation of β-CH_2_ protons probably occurs also due to the modulation of the hfi tensor by intramolecular rotations around the aliphatic bonds. These rotations can have low activation energy; so, in a small range of temperatures, the rotational frequency changes weakly, which causes the apparent independence of the relaxation time T_1_ on the temperature.

## 3. Discussion

### 3.1. Comparison of the Calculations and Experiment

There are numerous comparisons between the results of electron transfer theory and experiments [81,82]. We discuss here the comparison of the electron transfer reaction rate constants calculated from the Marcus cross-relation and those found in the experiments, as shown in Table 7. The experimental rate constants were corrected for diffusion using Equation (30).

Rate constants were calculated using Equation (34) with parameters from Table 5. All experimental rate constants were measured at 25 °C. We assume that parameters of the dipeptide Gly-TyrO^−^ do not differ from those of N-AcTyrO^−^ and parameters of the amide N-acetyltyrosine N-AcTyrO^−^-NH_2_ differ from those of N-AcTyrO^−^ only in charge.

The agreement with the experiment for reactions involving GMPH^+^/GMPH^•++^ and GMP(-H)^−^/GMP(-H)^•^ is noticeably worse than for reactions involving TyrO^−^/TyrO^•^ and N-AcTyrO^−^/N-AcTyrO^•^. When GMP is an oxidizing agent, the calculated rate value is overestimated; when GMP is a reducing agent, the calculated value is underestimated. Perhaps the values of standard potentials for GMPH^+^/GMPH^•++^ and GMP(-H)^−^/GMP(-H)^•^ are overestimated; if the error is in the pre-exponent or in the reorganization energy of GMP, the values would be overestimated or underestimated in all reactions involving GMP, regardless of whether it involves oxidation or reduction.

For the reactions of TyrO^−^ + GMP(-H)^•^ and N-AcTyrO^−^ + GMP(-H)^•^, the calculated rate values notably differ from the experimental values more than in other GMP reactions. This difference can be explained not only by the overestimation of the potential of GMP(-H)^−/•^ but also by other causes.

For the reactions involving TyrO^−^/TyrO^•^ and N-AcTyrO^−^/N-AcTyrO^•^, the calculated values agree well with the experimental values. In the case of the reaction of TyrO^−^ + N_3_^•^, the experimental rate constant is close to the diffusion rate constant; the calculation shows that the reaction occurs in the diffusion control limit and shows good agreement with the experiment.

### 3.2. The Possible Reasons for the Failure of the Marcus Cross-Relation in the Reactions of GMP(-H)^•^ Radical Reduction by TyrO^−^ and N-AcTyrO^−^ Anions

The Marcus cross-relation is fulfilled under the following conditions: the value of ΔG_12_^0^ is not too large, the reaction is far from the inverted region, the reactants and products are in the ground state, and the electron transfer is adiabatic [1,92]. Also, the size difference of the reactants should not be very large (2λ_12_ ≈ λ_11_ + λ_22_ requires 2(r_11_r_22_)^1/2^ ≈ r_11_ + r_22_); otherwise, the expression for the cross-relation should be modified [93].

The Marcus cross-relation for reactions of transition metal complexes was verified in [94,95,96] and for reactions of organic molecules in [97,98]. The Marcus cross-relation for redox reactions between organic TMPPD and various inorganic oxidating ions was tested in [99]. A good linear relation over a range of seven orders of magnitudes was found. As a rule, the calculations according to the cross-relation reaction rate constant turn out to be greater than the experimental ones; the error increases with the growth of K_12_. In the above-mentioned works, verification of the cross-relation was performed for rate constants measured only at one temperature, with the employment of Equations (2)–(4). In these equations, the parameter ln(f_12_) = −(ΔG_12_^0^)^2^/(2RTλ_12_), with f_12_ ≈ 1 at ΔG_12_^0^ << λ_12_. However, at large values of ΔG_12_^0^, this parameter contributes significantly, and its calculation requires values of the pre-exponents for DEE reactions. In these works, estimates in the order of 10^11^ were taken for the values of the pre-exponents. We believe that this can introduce an error when K_12_ is large. In this work, the values of the pre-exponents and reorganization energies of DEE were determined experimentally; this allowed us to exclude this error. In particular, it turned out that the pre-exponent in the case of GMP was two orders of magnitude greater than 10^11^ M^−1^s^−1^.

In the course of reactions, no isomer product was formed since, as seen from the CIDNP spectra, the electron was transferred only within the aromatic π-systems of the reactants.

When describing the kinetics of electron transfer, the pre-exponent includes the factor κ, which has the meaning of the probability of electron transfer while passing into the transition state. For adiabatic reactions κ = 1; also, the parameter κ is close to unity if there is substantial overlap of the reactant orbitals [100]. For reactions of organic molecules, κ = 1 is usually assumed [97,101]. In addition, possible nonadiabaticity is taken into account in the cross-relation, as κ_12_ = (κ_11_κ_22_)^1/2^.

The steric cross-reacting factor is included in the pre-exponent A_12_ and is estimated as the geometric mean of the steric factors of the DEE, i.e., A_12_ = (A_11_A_22_)^1/2^.

Another possible reason for the failure of the Marcus cross-relation is the effect of non-electrostatic interactions between the reactants. In the pairs TyrO^−^/TyrO^•^, GMP(-H)^−^/GMP(-H)^•^, and TyrO^−^/GMP(-H)^•^, the stacking interactions increasing the stability of pre-reaction complexes is possible. Assume that contribution ΔH^0^ of the stacking interaction to the enthalpy change of the pre-reaction complex formation is independent of temperature and ΔH^0^ < 0. Then, the expressions for the DEE and cross-reaction rate constants, taking into account the stacking interaction, are as follows:(45)$$K_{A}=K_{0}exp\left(\frac{-\Delta H^{0}-w_{r}}{RT}\right)$$
(46)$$k_{11}=K_{A}^{11}k_{ex}=\frac{A_{11}}{\sqrt{T}}exp\left(\frac{-\Delta {H_{11}}^{0}-w_{r}}{RT}\right)exp\left(\frac{-\lambda_{11}}{4RT}\right)$$
(47)$$k_{calc}^{red}=\frac{\sqrt{A_{11}A_{22}}}{\sqrt{T}}exp\left(\frac{-\Delta {H_{12}}^{0}-w_{12}}{RT}\right)exp\left(-\frac{{\left(\Delta {G_{12}}^{0}-w_{12}+w_{21}+\left(\lambda_{11}+\lambda_{22}\right)/2\right)}^{2}}{2RT\left(\lambda_{11}+\lambda_{22}\right)}\right)$$

In Equation (47), we assume that ΔH_12_^0^ ≈ ΔH_21_^0^. Equation (46) shows that when treating the temperature dependence of the DEE rate constants using the method described above (i.e., considering only the electrostatic interactions), the found values of the reorganization energies λ_11_^obs^ will be underestimated: λ_11_^obs^ = λ_11_ + 4ΔH_11_^0^.

Consideration of the stacking interaction on the one hand increases the calculated cross-reaction rate constant $k_{calc}^{red}$ due to the factor exp(−ΔH_12_^0^/RT); on the other hand, it decreases as the calculated reorganization energy increases. The final result depends on the values of ΔH_11_^0^, ΔH_22_^0^, and ΔH_12_^0^. When ΔH_11_^0^ = ΔH_22_^0^ = ΔH_12_^0^ = −RT, the calculated value of $k_{calc}^{red}$increases 1.4 times; when ΔH_11_^0^ = ΔH_22_^0^ = -RT, ΔH_12_^0^ = 0, it decreases 2 times.

The Marcus cross-relation fails if the electron transfer reaction produces products in the vibrationally excited state.

When we assume that the overlap of the orbitals in the TyrO^−^/GMP(-H)^•^ pair is less than in the TyrO^−^/TyrO^•^ and GMP(-H)^−^/GMP(-H)^•^ pairs, then the electronic factor of the cross-reaction appears to be less than that calculated from the cross-relation, κ_12_ < (κ_11_κ_22_)^1/2^. The enthalpy of the stacking interaction for the cross-reaction turns out to be lower than for the DEE reactions. This gives rise to the fact that the cross-reaction rate constant is less than the one we calculated without taking these effects into account.

## 4. Materials and Methods

The setup for the time-resolved CIDNP experiments was based on a Bruker DRX-200 NMR spectrometer (Bruker Corporation, Billerica, MA, U.S.; magnetic field 4.7 Tesla, resonance frequency of protons 200 MHz). A detailed scheme of the setup and methodology of the experiment was published in a review devoted to the application of time-resolved CIDNP for studying the kinetics and mechanism of reactions of biologically important molecules [102].

The ^1^H CIDNP spectra were recorded as follows. First, the sample in a standard 5 mm NMR Pyrex ampule was barbotaged with argon for 7 min to remove dissolved oxygen. Then, the sample was placed in the NMR spectrometer sensor, and broadband homonuclear de-coupler pulses were applied for a few seconds on channel ^1^H to suppress the thermal magnetization of the sample. The pulse sequence WALTZ16 was used. The parameters of the sequence were chosen so that the ^1^H NMR signals were completely absent from the spectrum without laser irradiation. After that, the sample in the ampoule was irradiated with a single laser pulse of the excimer XeCl-laser COMPEX Lambda Physik (Lambda Physik AG, Göttingen, Germany; wavelength 308 nm, pulse energy up to 120 mJ, pulse duration ~15 ns), and after time interval τ, a registering radiofrequency (RF) pulse with a maximum allowable power of −4 dB and a duration of 1 or 2 μs was applied. After the pulse was applied, the decay of the free induction signal was recorded using the same method as in the conventional NMR experiment. The delay τ was varied in the range from 0 to 100 µs.

The temperature was calibrated using the difference of chemical shifts in the ^1^H NMR spectrum of methanol; the temperature measurement error was 1 K [103].

The pH of the NMR samples was adjusted by the addition of NaOD. No correction was made for the deuterium isotope effect on the pH. Experiments with TyrO^-^ and N-AcTyrO^−^ were performed at pH = 11.7, with GMP(-H)^−^ and GMP(-H)^−^ + TyrO^−^ at pH = 11.3. At t = 25 °C, the pK_a_ of phenol groups TyrOH and N-AcTyrOH were 10.1 and 10.2 [104], pK_a_(GMP) = 9.4 [66], and pK_a_(GMP(-H)^•^) = 10.8 [67], but deprotonation was slow, and the GMP(-H)^−^ radical was stable at pH = 11.8 for at least 1000 µs [27]. The 2,2′-dipyridyl (DP) was used as a dye (photosensitizer); its concentration was 15 mM in all experiments. The concentrations of quencher C_q_ were chosen so that the characteristic quenching time k_q_^−1^C_q_^−1^ was shorter than the duration of the recording RF pulse (1–2 μs) and so that a rapid decline in polarization due to too high a product of the observed DEE rate constant and the quencher concentration, k_obs_C_q_, was avoided. The compositions of all samples are given in the Appendix A.

The GMP in the form of sodium salt hydrate (mass fraction of water of 23.4% as determined by ^1^H NMR), N-AcTyrOH, TyrOH, and D_2_O from “Sigma-Aldrich, St. Louis, MO, USA” were used without additional purification; DP was recrystallized from hexane. For the experiment, GMP(-H)^−^ + DP was tested at 65 °C; d8-DP was used because the signals from DP and GMP overlapped.

At each temperature, two or three kinetic curves differing in the quencher concentration were recorded. To record each kinetic curve, 4 samples were used for the GMP(-H)^−^ + DP, N-AcTyrO^−^ + DP, and TyrO^−^ + DP systems, and 8 samples for the GMP(-H)^−^ + TyrO^−^ + DP system. Each sample was used twice; the delays were followed in increasing order and in decreasing order to reduce the error from depletion. At each time delay, 4 scans (6 for TyrO^−^ + DP at 55 °C and 65 °C) were taken, so that each point of the kinetic curve corresponds to 64 signal accumulations for GMP(-H)^−^ + TyrO^−^ + DP, 48 signal accumulations for TyrO^−^ + DP at T = 55 °C and 65 °C, and 32 signal accumulations for GMP(-H)^−^ + DP, N-AcTyrO^−^ + DP, and TyrO^−^ + DP at T = 8–45 °C. A 2 μs RF pulse was used for GMP(-H)^−^ + TyrO^−^ + DP and TyrO^−^ + DP at T = 55 °C and 65 °C; otherwise, a 1 μs RF pulse was used.

All standard electrode potentials, E^0^, are reported versus NHE.

## 5. Conclusions

The CIDNP technique with microsecond time resolution makes it possible to study the kinetics of DEE reactions for the short-lived radicals and determine the DEE reaction rate constant by performing the reaction at different concentrations of the diamagnetic reactant.

The degenerate electron exchange reactions of the neutral radicals GMP(-H)^•^, TyrO^•^, and N-AcTyrO^•^ with anions GMP(-H)^−^, TyrO^−^, and N-AcTyrO^−^ in alkaline aqueous media were studied by CIDNP in the temperature range of 8–65 °C.

At each temperature chosen in this range, all parameters of CIDNP kinetics were determined, and the experimental values of reorganization energies for the DEE rate constants were found. At t = 25 °C, the rate constant of GMP(-H)^•^ reduction by the TyrO^−^ anion was measured.

The agreement of the calculated rate constants employing the Marcus cross-relation approach with the experimental ones for the reactions involving GMPH^+^/GMPH^•++^ and GMP(-H)^−^/GMP(-H)^•^ is poor; perhaps overestimated values of the standard electrode potentials for GMPH^+^/GMPH^•++^ and GMP(-H)^−^/GMP(-H)^•^ were used.

The rate constants of GMP(-H)^•^ radical reduction by tyrosine and N-acetyltyrosine anions calculated from the cross-relation differ by almost two orders of magnitude from those found experimentally, which is a notably greater difference than that for the other reactions involving GMP(-H)^−^/GMP(-H)^•^ species. Possible causes for this difference are not taking into account the nonadiabaticity of the cross-reaction [105] and the enthalpy of the stacking interaction between the reactants in the calculation.

At the same time, the calculated rate constants according to the cross-relation of electron transfer reactions involving TyrO^−^/TyrO^•^ and N-AcTyrO^−^/N-AcTyrO^•^ coincide well with the literature values.

It was also found that the dependences of the nuclear paramagnetic relaxation rate T_1_ on temperature are described by the Arrhenius dependence; for the methylene protons of tyrosine and N-acetyltyrosine radicals TyrO^•^ and N-AcTyrO^•^, these dependencies are almost activationless, while for the N-AcTyrO^•^ 3,5 protons, GMP(-H)^•^ 8th proton, and 3,4 protons of DPH^•^, the activation energy value coincides with the solvent (D_2_O) activation energy. This indicates a difference in relaxation mechanisms for these protons; relaxation of the methylene protons of the TyrO^•^ and N-AcTyrO^•^ radicals occurs due to the modulation of the hfi tensor due to ring inversion, while relaxation in the 3,5 protons of N-AcTyrO^•^ and 8th proton of the GMP(-H)^•^ occurs due to modulation of the hfi tensor as a result of stochastic rotation of the molecule as a whole. The intermediate type of the temperature dependence of the nuclear paramagnetic relaxation rate T_1_ is observed for the 3,5 protons of the TyrO^•^ radical; the activation energy was about half the activation energy for the solvent viscosity. We assume that the both relaxation mechanisms manifest themselves in this case.

## Data Availability

Not applicable.

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
