# Peer review of "Marcus Cross-Relationship Probed by Time-Resolved CIDNP"

_ijms, 2023, doi:10.3390/ijms241813860_

Round 1
Reviewer 1 Report
This is an interesting application of the time-resolved CIDNP method to reaction kinetics in biological systems. Of particular interest is the unique ability to measure degenerate chemical reactions such as electron transfer (DEE). Additionally, the measurement of DEE is a way of direct measurement of re-organization energy (lambda) because the activation energy of the reaction is equal to the re-organization energy. As I know, only TR-CIDNP or time-resolved ESR line shape analysis could be considerable method for such measurements. In this study, they have accurately measured DEE by TR-CIDNP and discussed the Marcus equation for cross-reaction rates. One of my concerns is that more than two kinetic parameters (k and T1) can be obtained from a time profile consisting of about 10 data points, and the validity issue is whether the correction for molecular diffusion is accurate. Another concern is that in the reaction systems used in Table 7, the DEE rates for GMP and Tyr derivatives are based on the DEE rates obtained in this study, but all other data are from references, and most of the k12 data are from other papers. Therefore, the reviewer a little doubt the data is enough reliable to test the Marcus cross reaction formula especially the validity if one can use the average of the two re-organization energies for DEEs as that for the cross reaction.
However, the reviewer thinks that it is worth to publish in the Journal.
Minor correction: The square is missing on the shoulder of exponential (lambda+deltaG0) in eq.(1).
Font size of a),b),c),d) are different in the same figure.
Author Response
Reviewer 1
We are very grateful to the Reviewer 1 for useful suggestions, comments and evaluation on our manuscript. We did all corrections, additions and insertions the Referee has demanded. Besides, presentation of the figures was improved; for covenience, Figure 8 was moved to the beginning and renamed to Figure 1. Figure 4 was renamed to Scheme 1. All figures were renumbered accordingly.
This is an interesting application of the time-resolved CIDNP method to reaction kinetics in biological systems. Of particular interest is the unique ability to measure degenerate chemical reactions such as electron transfer (DEE). Additionally, the measurement of DEE is a way of direct measurement of re-organization energy (lambda) because the activation energy of the reaction is equal to the re-organization energy. As I know, only TR-CIDNP or time-resolved ESR line shape analysis could be considerable method for such measurements. In this study, they have accurately measured DEE by TR-CIDNP and discussed the Marcus equation for cross-reaction rates.
One of my concerns is that more than two kinetic parameters (k and T1) can be obtained from a time profile consisting of about 10 data points, and the validity issue is whether the correction for molecular diffusion is accurate.
Replay:
At each temperature, two series of measurements were made with radical concentrations differing by a factor of two. Their fitting was carried out with common parameters k and T1. Note that the nuclear longitudinal relaxation time in radicals, T1, does not depend on the radical concentration. We believe that this allows us to reliably determine both parameters (k and T1). Further, considering that the constants k in all cases are smaller than the diffusion constant, the correction for molecular diffusion by equation (30) is correct.
Another concern is that in the reaction systems used in Table 7, the DEE rates for GMP and Tyr derivatives are based on the DEE rates obtained in this study, but all other data are from references, and most of the k12 data are from other papers. Therefore, the reviewer a little doubt the data is enough reliable to test the Marcus cross reaction formula especially the validity if one can use the average of the two re-organization energies for DEEs as that for the cross reaction.
Replay:
We agree with the Reviewer that in Table 7 only one-third of the data for DEE rate constants are taken from our previous work. We believe that the k12 data from other works in Table 7 are sufficiently reliable to verify the Marcus cross-reaction, there is no sufficient reason not to trust them. At the same time, we agree that the use of the average of two reorganization energies for DEE is an approximation.
However, the reviewer thinks that it is worth to publish in the Journal.
Replay:
Thank you for the kind recommendation.
Minor correction: The square is missing on the shoulder of exponential (lambda+deltaG0) in eq.(1).
Replay:
Correction is done.
Font size of a),b),c),d) are different in the same figure.
Replay:
Correction is done.
Reviewer 2 Report
1. The manuscript contains rather exhaustive numbers of equations, which seems to be copied from the previous papers. I think your own explanations are needed. For an example, equation (6) can be rewritten as
dP_R/dt = -k_sum*( P_R - P_R,eq)
where,
k_sum = k_R*R + k_obs*C_q + 1/T_1
P_R,eq = (-1)*k_R*beta*R^2/k_sum.
This is similar to the following Bloch equation for the z magnetization recovery.
dMz/dt = (-1)/T_1*(Mz - Mz,eq)
Why the equilibrium polarization term P_R,eq exist?
2. I would like to see more detailed derivation of equations (21) and (22). The limiting case w_r -> 0, is derived by P. W. Atkins in Physical Chemistry.
3. In Sec. 3.2, you discusses why Marcus cross-relation fails. The following papers also discuss the same subjects.
P. F. Barbara, et al., J. Phys Chem. 100, 13148 (1996).
E. J. Piechota and G. J. Meyer, J. Chem. Educ. 96, 2450 (2019).
More recently, JCP Special Topic on 65 years of electron transfer was published.
C.-P. Hsu, L. Hammarstrom and M. D. Newton, JCP 157, 020401 (2022).
I think some more reading may improve your discussion.
Author Response
Reviewer 2
We are very grateful to the Reviewer 2 for useful suggestions, comments and evaluation on our manuscript. We did all corrections, additions and insertions the Referees have demanded. Besides, presentation of the figures was improved; for covenience, Figure 8 was moved to the beginning and renamed to Figure 1. Figure 4 was renamed to Scheme 1. All figures were renumbered accordingly.
- The manuscript contains rather exhaustive numbers of equations, which seems to be copied from the previous papers. I think your own explanations are needed. For an example, equation (6) can be rewritten as
dP_R/dt = -k_sum*( P_R - P_R,eq)
where,
k_sum = k_R*R + k_obs*C_q + 1/T_1
P_R,eq = (-1)*k_R*beta*R^2/k_sum.; (6)
This is similar to the following Bloch equation for the z magnetization recovery.
dMz/dt = (-1)/T_1*(Mz - Mz,eq)
Why the equilibrium polarization term P_R,eq exist?
Repay:
There are two types of radical pairs. The first type is electron-spin correlated pairs formed in the primary act of photoinduced electron transfer called geminate pairs. A geminate CIDNP is a CIDNP formed by recombination of these pairs. Radicals that have not recombined and escaped into the bulk of solution can meet with the another such radicals and form the so-called F-pair (Free pair), the state of electron spins of which is not correlated at the first contact of radicals. Nevertheless, these pairs in the recombination process also generate nonequilibrium nuclear polarization in diamagnetic recombination products, F-pair CIDNP. The term P_R,eq = (-1)*k_R*beta*R^2/k_sum in Reviewer notations is precisely responsible for the CIDNP generated in F-pair recombination and is proportional to the radical concentration quadratic
- I would like to see more detailed derivation of equations (21) and (22). The limiting case w_r -> 0, is derived by P. W. Atkins in Physical Chemistry.
Replay:
We added two references where the detailed derivation of equations (21) and (22) is done:
- a) P. Debye, Trans. Electrochem. Soc. 82, 265-272 (1942).
b) R. M. Noyes, Prog. React. Kinet. 1, 129-160 (1961).
In Sec. 3.2, you discusses why Marcus cross-relation fails. The following papers also discuss the same subjects.
- F. Barbara, et al., J. Phys Chem. 100, 13148 (1996).
- J. Piechota and G. J. Meyer, J. Chem. Educ. 96, 2450 (2019).
More recently, JCP Special Topic on 65 years of electron transfer was published.
C.-P. Hsu, L. Hammarstrom and M. D. Newton, JCP 157, 020401 (2022).
- J. Piechota and G. J. Meyer, J. Chem. Educ. 96, 2450 (2019).
I think some more reading may improve your discussion.
Replay:
We have taken all these publications into account in the manuscript.